# Item Response Theory Analysis of the Dark Factor of Personality Scale for College Students in China

**DOI:** 10.3390/ijerph191912787

**Published:** 2022-10-06

**Authors:** Xinyi Wang, Shiyi Zhang, Tao Xin

**Affiliations:** Collaborative Innovation Center of Assessment for Basic Education Quality, Beijing Normal University, Beijing 100875, China

**Keywords:** the dark factor of personality scale, psychometric properties, item response theory

## Abstract

The Dark Factor of Personality (D) describes the common core of dark traits and is a stable indicator for socially aversive behaviors. This study investigated the psychometric properties of the Chinese version of the Dark Factor of Personality Scale for college students using item response theory (IRT). A total of 762 students—251 males and 511 females (*M* = 19.99, *SD* = 1.30)—were recruited. Item response theory methods were utilized to evaluate the properties of the scale. Four items with poor item properties were excluded, obtaining a final 28-item scale (D28-C) that included highly discriminative items showing high measurement precision in various levels of the D factor. Furthermore, a test of differential item functioning (DIF) by gender was conducted. The result indicated that the scale as a whole could be seen as gender invariant. Lastly, according to the detailed information provided by IRT and the content of items, a reliable short form of the D28-C comprising 15 items was obtained. The study enriched the existing knowledge of the dark factor of personality in the Chinese background and made some revisions to the corresponding scale to make it a more reliable tool for measurement in China. In addition, the shortened version of the scale based on item information and content helps to improve the efficiency of the measurement.

## 1. Introduction

Dark traits, stable personality dispositions associated with ethically and socially aversive behaviors, have recently received more attention in personality psychology recently [1]. The most prominent theory about dark traits is “Dark Triad” proposed by Paulhus and Williams [2]. Dark Triad has three components: Machiavellianism, Narcissism, and Psychopathy. In recent years, some other personality traits with dark aspects, which means that these traits are always associated with negative outcomes, were suggested to be included in the framework of dark traits, such as spitefulness and greediness [3]. It is also found that there are conceptual and empirical overlaps among these different aversive traits [1]. We can find that both the “dark aspects” and the overlaps in the previous studies indicate commonalities with the aversive traits. To describe and explain the underlying commonalities, Moshagen and his colleagues proposed the theory of the Dark Factor of Personality (D) and regarded D as a general disposition of all aversive traits rather than a specific dark trait [1].

Dark traits are often regarded as representatives of the dark side of the personality, are connected with negative outcomes, and the connections are cross-culturally stable. Many studies from Western countries have found that scores on the Dark Triad characteristics are positively associated with aggression, bullying, and cheating [4,5,6]. Results from various studies conducted in China also indicated that aggression is a distinctive characteristic outcome of the Dark Triad [7]. One study on Chinese university students found that Dark Triad constructs can positively predict scholastic cheating [8]. However, despite the negative influences mentioned above, some characteristic behaviors of dark traits are also seen as adaptive survival strategies which are helpful for individuals to adapt to the environment [9]. It is also found that some dark traits are beneficial in the competitive environment, and high scores on some dark traits are positively correlated with success [10]. For example, individuals with a high level of Machiavellianism are good at handling various tasks flexibly and leading others in the workplace [11]. A study conducted in Germany also showed that Machiavellianism was positively associated with leadership position and career satisfaction [12]. Therefore, studying the dark traits and their relationships with psychological or behavioral outcomes will not only help to understand and intervene in socially aversive behaviors, but it may also be helpful in everyday areas, such as the company’s recruitment and selection process of positions requiring leadership. To conduct relevant research on dark traits, a precise and valid tool for measuring the traits is necessary.

Among all measures, the Dark Factor of Personality Scale is a relatively new scale based on the Dark Factor of Personality theory. Three versions of the Dark Factors of Personality Scale were developed to measure D, and they, respectively, included 70, 35, and 16 items (thus donated D70, D35, and D16) [13]. All three versions had good reliability and validity and it was also verified that they performed well in some Western countries (e.g., Germany) [13,14]. Although the scales have been used in the Western context in several studies, no studies on D have been conducted in Chinese samples. However, given that there are often cultural differences between Western and Eastern countries, the definition of dark traits may vary in East countries [11]. Thus, it is necessary to verify whether the scales are applicable to the sample in China. To compensate for this limitation, we created and evaluated a Chinese version of the D scale.

Additionally, previous studies about the psychometric properties of D scale are all based on Classic Test Theory (CTT). However, it is found that CTT has some shortcomings, such as sample dependence, which may negatively influence the estimation results. By contrast, item response theory (IRT) provides a more comprehensive description of psychometric properties of scales at both the item and scale level than traditional psychometric methods based on the classical test theory (CTT). IRT methods can also provide item and test information, which can help to discriminate which items are more attributive to the whole scale. It is particularly useful because researchers can remove ineffective items to shorten the scales and improve the efficiency of the scale.

Thus, the purpose of this study is: (1) to translate the Dark Factor of Personality Scale into Chinese and utilize IRT to evaluate the psychometric properties of the Chinese version scale; and (2) to develop a short version of the Chinese D scale to facilitate a quick and efficient assessment. The present study can examine whether the scale is appropriate in China and provide additional psychometric information about the scale that cannot be obtained using classical psychometric methods.

## 2. Background

### 2.1. The Theory of the Dark Factor of Personality

One of the most widely known theories about dark traits is the Dark Triad, comprising Machiavellianism, Narcissism, and Psychopathy [2]. Machiavellianism is characterized by utilizing others for one’s own profit; Narcissism is characterized by a set of cognitive styles (e.g., grandiosity, entitlement, dominance, and superiority); and characteristics of Psychopathy include high levels of impulsivity and low levels of empathy and anxiety [2,15]. Scores on Dark Triad are regarded as significant predictor variables for numerous negative psychosocial outcomes (e.g., interpersonal difficulties) and socially aversive behaviors (e.g., financial cheating) [6,16]. The theory has been widely cited since its construction and has contributed to the tremendous growth of the field [17], and a variety of scales have been developed based on this theory [18,19]. With a growing number of different dark traits being introduced, it was found that there are conceptual and empirical overlaps among these aversive traits [1]. What’s more, one result of multiple dark traits is that if researchers want to measure an individual’s level of dark traits, they need as many scales of different dark traits as possible to ensure the representativeness and validity of the measurement results. However, having too many scales and items would reduce the efficiency of the measurement. To describe the common core of all dark traits and their relationships with other traits and behaviors, the Dark Factor of Personality (D) was proposed [1].

D was defined as “the general tendency to maximize one’s individual utility—disregarding, accepting, or malevolently provoking disutility for others—accompanied by beliefs that serve as justifications”. The individual utility is related to psychological or monetary or status-striving goals, and disutility for others refers to any types of costs, such as material cost and emotional cost [1]. It is worthy to note that D is a general disposition of all aversive traits rather than a specific dark trait. From the psychometric perspective, D is a general factor that can describe the overlap of all dark traits, whereas the remaining parts of each trait will be described by specific factors. It is the specific factors that allow individuals with an equivalent level of D to have different behavioral characteristics. For example, individuals who score high on Narcissism may be more interested in getting attention from others, while individuals who score high on Machiavellianism may be better at using strategies to achieve goals. The framework of the Dark Factor of Personality has gotten some empirical support. In a series of studies, Moshagen and his colleagues extracted a general factor from a set of dark traits (e.g., Egoism, Moral Disengagement, and Spitefulness, etc.) with a bifactor model, which is in accordance with the concept of D. The results of the research showed that bifactor model was stable in a 4-year longitudinal study and the D factor can effectively predict dark traits and behavioral outcomes. For example, it can significantly and positively predict the internalized moral identity and negatively predict aggression [1].

### 2.2. Measuring the Dark Factor of Personality

D is regarded as a fluid construct, meaning that it is a common core shared by all aversive traits, and it cannot be represented well by a specific dark trait. Its comprehensive component of personality is helpful in exploring more accurate connections between dark personality traits and socially aversive behaviors [10]. To have a reliable measurement of D, it is necessary to cover a sufficient number of dark personality traits. Consequently, an item pool comprising 12 scales for different dark traits was developed. With a series of item selection procedures applied to the item pool, Moshagen and his colleagues developed three versions of the Dark Factors of Personality Scale (D70, D35, and D16) for the measurement of D [13].

D70 is a bifactor scale with one general factor and six specific factors. It includes 70 items of which 34 are negatively keyed. D35 and D16 are both short versions of D70, and they are single-factor models composed of 35 items (17 negatively keyed items) and 16 items (8 negatively keyed items), respectively [13]. It is shown that they can provide a reliable measurement of D. The internal consistency is high (Cronbach’s α = 0.91–0.97), and the retest reliability over a 34-period is also high (*r*_tt_ = 0.90–0.95) [13]. The scales also appeared to have cross-cultural consistency and stability. Researchers applied all three versions of the scales to the German population and found that the German versions of the D70, D35, and D16 had the same structures as the original version and performed well on the indexes of internal consistency (Cronbach’s α = 0.80–0.93). The result also showed that D was positively related to socially undesirable behaviors such as violence, at a significant level [14]. After the scales were developed, they had also been widely applied in empirical research within a short period. For example, it was applied to Spanish subjects to explore the relationships between dark personality traits and the use of online dating software [20]. It was also applied to explore self-regulation behaviors in Danish populations in the context of COVID-19 [21].

Although some researchers translated the D scales into distinct languages and applied them to practical studies, they were all conducted in Western countries. There is little information about whether the construct and definition of the Dark Factor of Personality are also the same in the world’s largest country, China. Although there have been some Chinese versions of scales measuring levels of dark traits (e.g., SD3-C) [22], it is still necessary to develop a Chinese version of the D factor scale.

Firstly, given the cultural differences between East and West, the definition and structure of D may be different in Chinese cultural contexts. Some researchers pointed out that the Dark Triad is more compatible with Western culture and is an extreme development of Western individualism that emphasizes competition, whereas it is far from the cooperation and sacrifice emphasized by collectivism in Eastern countries [11]. It is also found that individuals from the US have higher scores on Narcissism than individuals from China [23]. Therefore, there may be differences in individuals’ definitions of dark traits in the East, which may affect the validity of the measurement of D. Meanwhile, previous studies have found that the Short Dark Triad, a widely used scale based on the theory of the Dark Triad, had a poor fit to the three-factor model and unsatisfactory reliability after it was translated into Chinese, indicating that the scale may not be applicable in the Chinese cultural context [24]. Thus, previous studies showed that there might be cultural differences in dark traits from theoretical and practical perspectives, reminding us that people from different cultural backgrounds may comprehend the dark factor of personality differently. Therefore, verifying the structure of D and the corresponding scales in the Chinese context is necessary. Secondly, the Chinese version of the scales available for measuring dark traits are mostly based on the Dark Triad theory (e.g., SD3-C and DD-C) or scales measuring specific dark traits (e.g., NPI13-C) [22,24,25], and there is no scale based on the theory of the Dark Factor of Personality. As a relatively new theory of dark personality traits, it focuses on the common core of all dark traits rather than a specific trait. It was also believed that scales based on this theory could help to predict the associations between dark traits and the relevant outcomings (e.g., socially aversive behaviors) [10].

It is found that most studies regarding the psychometric properties of the Dark Factor scales (both in English and non-English) were based on the classical test theory (CTT). CTT uses the concept of “true score” to present an individual level of a specific trait or ability. It is an invariant value comprising the observed score (the score of the scale) and random error. Although CTT is easy to understand and widely used, it has shortcomings that may bring about negative influences. Some of its assumptions are always violated in empirical studies, such as the one that takes standard error of measurement as uniform for all scores in a particular population [26]. Moreover, the analysis methods are sample-dependent which may produce biased results if the scales were to be applied to a different sample. To evaluate the scales more accurately and provide more reliable revision references, we should avoid the above problems that may affect the assessment results.

Among the psychometric theories, item response theory (IRT) can overcome these deficiencies, and it has also been widely used in personality, attitude, and ability measures [27]. The item parameters based on IRT are invariant across samples and do not rely on specific tests or items. Thus, the item parameters can be applied to different samples and scores are comparable across different tests. Furthermore, IRT uses the concept of information function instead of reliability, allowing for more accurate estimates of measurement error for items and scales at the individual level. Moreover, detailed information provided by IRT helps to remove items that offer little information to scales. Then, we can shorten the scale while retaining enough information for the scale.

What’s more, many studies have demonstrated that there is a gender difference in levels of dark traits. For example, several studies found that men had significantly higher scores on Machiavellianism, Narcissism, and Psychopathy than women, and this is also the same in China [24,28,29]. For the D factor, men also have higher mean scores than women [10,30]. It is also worth noting that, although stability across gender has been regarded as one of the item selection criteria when developing the D scales, the German version of D35 and D16 did not meet the strict criteria for invariance across gender [10]. Thus, it is essential to verify the DIF by gender of the Chinese version.

### 2.3. Research Goals

Given that the D35 can provide the items’ content integrity while not having too many items, which can improve the efficiency of personality measures, this study focused on the D35. Therefore, to verify the structure and quality of the Dark Factor scale in Chinese populations, the present study translated D35 into Chinese and examined the psychometric properties of the Chinese version of D35. Given the advantages of IRT over CTT and that no previous research has used IRT to analyze the D35, our study evaluated the Chinese version using an IRT model. A shortened scale was also obtained according to the detailed information provided by IRT methods.

## 3. Materials and Methods

### 3.1. Participants

Participants were recruited through flyers and advertisements distributed through social media and websites.

There were two samples included in the current study. Sample 1 was recruited to evaluate whether the Chinese translations of the items are understandable. It comprised 57 Chinese students between the ages of 17 and 22 years old (mean age = 19.70 years old, *SD* = 0.92; 77.32% female). Sample 2 was recruited for the formal examinations on the Chinese version of D scale using IRT methods. It comprised 762 Chinese students between the ages of 17 and 27 years old (mean age = 19.99 years old, *SD* = 1.30; 67.06% female). According to the recommended sample size in the previous research [31], the sample size can ensure the accuracy of parameter estimations. Participants’ demographics of Sample 2 are shown in Table 1.

### 3.2. Procedures

In the present study, four steps were conducted to get an initial Chinese version of D scale. First of all, with the permission of the original authors of D35, the questionnaire was translated into Chinese by a group of Chinese students majoring in psychology. Then, this version was back-translated by three fluent English speakers (an English major student, and two individuals living in the US) and sent to the original authors. By comparing the English version and with the advice of the original author, a group of three psychology students resolved the discrepancies and revised translations. Third, a pilot study of 57 college students was conducted to confirm whether the updated version presented any difficulties understanding or responding to it. At the same time, a psychology professor who is fluent in both Chinese and English was invited to evaluate the new version. With the results of the pilot study and the advice of the professor, 3 items that were difficult to understand and unsuitable in the context of Chinese culture and language customs were deleted. For example, item 31 (“For most things, there is a point of having enough.”) was chosen for measuring “greed” in the original scale. However, in the Chinese cultural context, greed is not always a trait that is negative (e.g., being greedy for knowledge). In addition, although greed has a negative meaning in some circumstances, it is not seen as a very serious sin in China as it is in Western culture [32]. Thus, it is inappropriate to take item 31 as an item for measuring dark traits. Finally, all researchers agreed to approve the revised version for use in the current study, obtaining a 32-item scale (the initial Chinese version of the D scale, D32-C).

A total of 762 participants signed up and took part in the study. The questionnaire started with the initial Chinese version of the D scale, followed by basic demographic characteristics. The order of items from the D scale was automatically randomized by the questionnaire platform to avoid potential influences of order. After reading and signing an informed consent form, participants were required to complete a series of questionnaires individually through the Tencent online survey platform. They would receive a report on D based on their responses to the questionnaire.

### 3.3. Measures

The Dark Factors of Personality Scale. The original D35 developed to assess the dark factor of personality is a self-report questionnaire comprising 35 items, including 17 negatively keyed items and 18 positively keyed items [13]. With the evaluation of an expert and students from the psychology apartment, 3 items were removed. As a result, the initial Chinese version used for measurement in the present study had 32 items with 15 negatively keyed items. Each item was rated on a 5-point Likert scale (1 = strongly disagree to 5 = strongly agree), with higher total scores indicating a stronger personality disposition to dark traits. In the current study, Cronbach’s α of the D32-C was 0.90.

### 3.4. Statistical Analysis

All the statistical analyses were completed with RStudio 1.4.1106. The analysis for IRT assumptions was conducted with the package “psych” (an R package that can provide multivariate analysis and basic analysis for scale construction like factor analysis) [33]; the parameter estimation and differential item functioning (DIF) were conducted with the package ”mirt” (an R package for analyzing dichotomous and polytomous data with latent trait models based on IRT) [34]; and the magnitude of DIF was conducted with the package ”lordif” (an R package which can detect DIF based on IRT) [35].

To develop a reliable Chinese version of the Dark Factor of Personality Scale, the analysis procedures in the present study were conducted as follows. Firstly, we verified the initial scale (32 items) with the IRT model. Before utilizing IRT methods, two IRT assumptions—unidimensionality and local dependence—were examined. Then, we chose the best fit IRT model from three alternative models and verified its fitness with *M*_2_ statistics and root-mean-square error of the approximation (RMSEA). After applying the IRT model to the data, we excluded items with poor discrimination and unsuitable difficulty, obtaining an updated scale. Secondly, we repeated the analysis procedures above for the updated scale to ensure that the new data could be analyzed with IRT methods. Then, item-level fit and DIF were detected on the updated scale. Item information and test information were also calculated as indicators of the reliability of the final version scale. The specific processes of the above analyses are shown below.

#### 3.4.1. Unidimensionality and Local Independence

Unidimensionality and local independence are two assumptions of statistical analysis using IRT models. In the present study, dimensionality was assessed using exploratory factor analysis (EFA). If the first factor explained more than 20% of the total variance and the ratio of the first and second eigenvalue was larger than 3, then the precondition of unidimensionality was acceptable [36,37]. Local independence means that individuals’ response to one item is not influenced by any other items [38]. In the present study, we used the Q_3_ index, residual correlations between items, to examine the local independence assumption. Local independence was accepted if more than 95% of the residual correlations for the items were smaller than 0.3 [39].

#### 3.4.2. Model Selection

Three polytomous IRT models—graded response model (GRM), generalized rating scale model (GRSM), and generalized partial credit model (GPCM)—were fitted to the data. To choose a more appropriate IRT model for more accurate results, we compared the three models based on three indices: the Akaike’s information criterion (AIC) values, Bayesian information criterion (BIC) values, and −2 × log-likelihood. Smaller values indicated a better-fitted model [38,40]. We used the *M*_2_ statistic and the associated RMSEA value as the criteria for the valuation of the model’s fitness. The fitness was good if the *M*_2_ statistic was not significant. Given that the significance of the *M*_2_ statistic is not stable among different sizes of samples, RMSEA was also considered [38]. If the RMSEA value was lower than 0.1, then it was an acceptable fit [41].

#### 3.4.3. Parameter Estimation

Under the GRM, discrimination (a) parameters and threshold (b_i_) parameters were obtained for each item with an EM algorithm approach. The discrimination parameter reflects the item’s capability of discriminating participants with different levels of the target trait. An ideal value of a parameter is larger than 1.0 and the items with values smaller than 0.75 were recommended for removal [42]. The threshold parameters indicate the difficulty of the items. When the threshold is large, only people with high levels of trait would choose the corresponding value. The common range of threshold values was −4 to 4 [43].

#### 3.4.4. Item Fit

To assess the fit of each item to the final model, *S*-*χ*^2^ item fit statistic was used [44]. A significant p indicates a bad fit on the item level. However, given that larger samples increase the possibility of statistical significance, the significance level of *S*-*χ*^2^ was adjusted by Bonferroni adjustment.

#### 3.4.5. Different Item Functioning

To examine whether female and male participants at the same trait level responded differently to the items, we examined the DIF by gender. The DIF magnitude was evaluated by the likelihood ratio *χ*^2^ test. Given that several comparisons were being conducted, we also used the Bonferroni adjustment to adjust the significance level [39]. Given that the statistical power is influenced by the sample size, the magnitude of the DIF was calculated. The pseudo *R*^2^ statistics were used as the index and when the value was smaller than 0.13, the detected DIF was negligible [45].

#### 3.4.6. Item and Test Information

The item information refers to the empirical information a specific item can provide across the entire range of trait levels and can be described by the item information function (IIF). However, given that the sample size of the extreme trait levels is often small which may lead to inflated information and error, the more commonly used range of latent trait level is (−3, 3) [46]. It reflects how much useful information the single item can provide to the total scale, and the items with rather low information are recommended for removal when developing the short form of the scales. Test information is the sum of the item information of all items, reflecting the reliability of the total scale. The higher test information at a trait level means that the scale can provide estimation for the participants of the level more precisely, and the scale is more reliable at the trait level. Thus, on the basis of item information and test information, it was possible to select items that contributed more information to the whole scale.

#### 3.4.7. Simplification of Scale 

To shorten the length of the scale while keeping the content of the questions as rich as possible, we adopted the method used by Meriac and his colleagues, selecting items based on the content source and item information [47]. Specifically, the RTI (reduction in test information if the corresponding item was removed) of each item was calculated first, and then the items were categorized according to the item source. In each category, half of the items with the highest RTI were retained to form the final short version of the scale.

## 4. Results

The initial Chinese version of the D scale consisted of 32 items. The mean score of each item is shown in Table 2. The total score of the initial scale for the whole students ranged from 38 to 147, and the overall mean total score was 79.78 (*SD* = 17.33). The mean total score difference between males (*M* = 81.06, *SD* = 17.73) and females (*M* = 79.15, *SD* = 17.11) was not significant (*t*_(760)_ = 1.43, *p* = 0.15, Cohen’s *d* = 0.11).

### 4.1. Unidimensionality and Local Independence

The KMO statistic was 0.93 and Bartlett’s test was statistically significant (*p* < 0.001), indicating that the data met the assumptions of EFA. The EFA results showed that the first factor explained 24% of the total variance, and the eigenvalues of the first and the second factor were 7.72 and 1.51, respectively, with a ratio larger than 3. Therefore, the 32-item scale could be regarded as unidimensional. It is worth noting that the two items (item 2 and item 16) had low loadings (less than 0.3) in the single-factor model, indicating that these items could be considered for removal in the later analysis. Moreover, we calculated the residual correlations of the items and the results showed that the absolute values of the correlations among residuals were all smaller than 0.3, indicating that the local independence assumption was met.

### 4.2. Model Selection

As shown in the Table 3, the GRM had the smallest values among the relative model-fit indices (−2LL, AIC, BIC), indicating that the GRM was more suitable for the initial Chinese version D scale data than GRSM and GPCM. Therefore, GRM was used for further analysis. Additionally, the fit statistics of the IRT model were calculated, and the results showed that the fit for the GRM was acceptable (*M*_2_ = 1462.23, *df* = 368, *p* < 0.001; RMSEA = 0.06, 95%CI [0.059, 0.066], TLI = 0.79, and CFI = 0.80).

### 4.3. Item Properties and Selection

The results of parameter estimations are presented in Table 4. The discrimination parameters ranged from 0.35 to 1.71. Most of the item discrimination values were larger than 0.75, except for items 2, 9, 15, and 16. Thus, these four items were also candidates for removal. For threshold parameters, all the items had ordered values with the first thresholds being the lowest, indicating that as the level of dark traits increased, the probability of responding to the items with higher scores was increasing. It is acceptable that there were some items with threshold values slightly larger than 4 or smaller than −4, but it is noteworthy that the first threshold parameter of item 9 was smaller than −5, and item 16 had a threshold value near 6, indicating that these items could be removed for their very low/high level of difficulty.

Based on the analysis above, we found that four items (item 2, 9, 15, and 16) did not perform well on the item properties. They had poor discrimination parameter values (≤0.75) or threshold parameter values that were far outside of the common range (≥4 or ≤−4). In addition, item 2 and item 16 had rather low loadings in the one-factor model. As a result, we removed items 2, 9, 15, and 16 and repeated the analysis procedures above for the updated, 28-item scale (D28-C).

### 4.4. Item Fit and Parameter Estimation

The 28-item scale still satisfied the unidimensionality and local independence assumptions of IRT, and the psychometric properties of each item were good. Table 5 presents the results of parameter estimations with GRM. The discrimination parameters ranged from 0.79 to 1.76, meaning that they all had moderate to high discrimination values. Thus, the D28-C had a good capability of discriminating students with different levels of dark trait disposition. For threshold parameters, all the items also had ordered values with the first thresholds being the lowest, indicating that the probability of choosing options with higher scores increases with the level of the D factor. Among all the items, item 21 had the smallest values for the third and fourth threshold values, and item 34 had the highest values for the last two thresholds. This means that item 21 and item 34 required the lowest and highest level of dark traits, respectively, to endorse options with high scores. Additionally, it is noteworthy that only the first threshold values were negative, and most of the second and third thresholds and all the fourth threshold values were positive. This means that most of the items in D28-C are “difficult”, namely, a higher level of D was needed to get a higher score.

### 4.5. Differential Item Functioning

DIF by gender was tested using the likelihood ratio *χ*^2^ test approach, and Table 6 presents the results of all the items. The *χ*^2^ values for DIF by gender ranged from 0.00 to 10.13. After Bonferroni adjustment, the results showed that all the items did not have severe DIF, except for item 13. Item 13 had the largest *χ*^2^ value (10.13) and it had significant DIF both before and after the Bonferroni adjustment. To confirm the magnitude of the detected DIF for item 13, the effect size was calculated. The result was 0.006, which is a negligible (<0.13) DIF according to the classification guideline of Zumbo [45]. In addition, because only 4% of the items in the D28-C were noninvariant, we determined that the D28-C was invariant as a whole [48].

### 4.6. Item and Test Information Function

Figure 1 displays the item information function for D28-C items. Information provided by items 1, 5, 12, 14, 20, 22, 34, and 35 was relatively greater across the continuum of dark trait disposition. Item 24 provided the largest amount of information (near 1) across the range of −1.0 to 3.0. Most items appeared to provide the largest amount of information for the students whose dark trait disposition was in the range of −1.0 to 2.0. In addition to the item information function, test information was calculated by summing up all the item information across the trait continuum ranging from −3.0 to 3.0. The total information of the test was 65.25, and Figure 2 describes the total information provided by D28-C. The solid line represents the test information curve and the dashed line represents the standard error of measurement for the whole scale. It shows that when the dark trait disposition continuum ranged from −0.73 to 2.72, the scale could produce slightly more test information (≥12.0) and less SE (≤0.30), which means that D28-C could provide a more precise measurement for the students whose dark trait disposition level fell in this range.

As a result, 15 items (items 1, 4, 7, 8, 10, 11, 13, 17, 22, 23, 24, 26, 28, 29, and 32) were retained and 6 items were negatively keyed. The IRT analysis results showed that the discrimination parameters ranged from 0.98 to 1.57 and the short form fit the GRM model well at the item level. The threshold parameters ranged from −3.59 to 5.13, with item 33 having the highest values for the last two thresholds, indicating that it was the most difficult item in the brief scale. Item information (across the trait level from −3 to 3) of the items ranged from 1.89 to 3.46, while the test information was 40.71, indicating that the short form retained 62.80% of the test information of the final version scale. As shown in Figure 3, the two curves had a similar trend, which means that they could both provide a more accurate measurement at some specific range of D levels than other ranges. The information curve of the brief scale was lower than that of D28-C, indicating that the removal of items did lead to some loss of the total information provided by the whole scale.

## 5. Discussions

This study examined the structure and reliability of D35 in the Chinese cultural context. It was found that the Chinese version scale was also unidimensional, but some of the items were not applicable in the Chinese context and, thus, were removed in the revision process. The final 28-item dark factor of personality scale provided a reliable instrument for measuring the D level in the Chinese-language samples. In addition, a short version of D28-C was developed based on the content and detailed information of the items, obtaining a 15-item scale.

After removing four items from the initial Chinese version of the scale, our Chinese version of the D scale (D28-C) was comparable to the original English version and had sound psychometric properties. EFA results showed that all items shared one latent structure, consistent with previous studies [15,20]. The discrimination parameters of the final scale ranged from 0.79 to 1.76, indicating that the items could discriminate different levels of D with adequate accuracy. For the four deleted items (item 2: “Payback needs to be quick and nasty.”; item 9: “Never tell anyone the real reason you did something unless it is useful to do so.”; item 15: “In principle, everyone is worth the same.”, and item 16: “I cannot imagine how being mean to others could ever be exciting. (R)”), their discrimination estimations were slightly small, indicating that it was difficult for these four items to distinguish among students with different levels of D factor. This discrepancy with the original scale could be explained from a cultural perspective. For items 2 and 16, it is generally known that collectivist culture encourages cooperation and tolerance and the Confucian culture also emphasizes benevolence [11], so behaviors that may hurt others are strongly inhibited in Chinese society. For item 9, it appears to emphasize strategy more than just dark traits such as Psychopathy. China has a long history of strategic culture and profoundly influences on many areas of individuals’ social life [49]. Being good at using strategies to achieve goals is not always negative in Chinese culture. Additionally, this item does not specify whether it is detrimental to the profit of others. Thus, it may not be highly related to dark traits, which weakens its capability to distinguish people with different levels of D. For item 15, unlike in Western culture, which emphasizes individual achievement, commonwealth and equality are more emphasized pursuits and consensuses in China’s collectivist culture [50]. Therefore, this item will be relatively more difficult to distinguish individual differences. Also, the high values of the third and fourth threshold parameter can provide some support for this reason.

The test of DIF identified nonsignificant DIF items after Bonferroni adjustment for gender, except for item 13 (“A person should use any and all means that are to his advantage, taking care of course, that others do not find out.”), which had a minor but significant DIF. According to the social role theory [51], the result is reasonable. Society and culture have different role expectations for different gender groups, which may influence the criteria to which individuals refer in their self-assessment. For instance, Machiavellianism is a dark trait marked by good uses of strategies. This trait characteristic is more in line with males’ instrumental roles (associated with work, achievement, and domination) and less with females’ expressive roles (associated with emotional expression and interpersonal relationships). Thus, using strategies to achieve goals is less encouraged for females, and they will be more sensitive about it. As a result, men with high levels may consider that they are at an average level. In contrast, women with average levels perceive themselves as being at a high level, resulting in the DIF across gender. However, given that item 13 accounted for only 4% of the whole scale and the effect size was small, we still concluded that the whole D28-C was gender invariant.

For the test information, the total information of the D28-C was 64.83 and the marginal reliability was 0.92, indicating that it was a reliable scale. It can be found that the information at the moderate and higher disposition levels (i.e., −1.35 < θ < 3) was over 10, which converted to reliability larger than 0.9, indicating sufficient measurement precision. Although the reliability dropped quickly out of the range of the continuum, it could still provide reasonable precision for most individuals at higher attitude levels. For instance, even for the low attitude level of −3.0, the information was still 4.26, which equaled acceptable marginal reliability of 0.77 [52]. Overall, the scale items provided acceptable test information for measuring the D factor of students, especially for those located in the moderate and higher range of the continuum, with high scale reliability in this interval.

Meanwhile, this study provided some interesting findings through the test information, which cannot be found with CTT methods. For example, it was found that the D scale had different measurement accuracy across the D continuum. With traditional methods, we can know only that the Cronbach’s α value was 0.91 for the final Chinese version of the scale, indicating that the scale had identical reliability for all D levels. However, the IRT analysis provided a more detailed explanation: the scale had differential measurement precision across the trait continuum. For example, the scale was highly reliable for individuals with moderate and higher trait levels and provided less, but acceptable, reliability for lower levels. Moreover, item information function displayed how the item information, a concept similar to the reliability in CTT, varied with the level of the D factor. From the function, researchers can discern which level range of the D is the most appropriate for this item. Then, in situations where it is necessary to differentiate among individuals with specific D levels, adding some “appropriate” items based on the item functions to the scale is recommended to improve accuracy at the required range of the trait continuum.

Though the 28-item scale was reliable, we want to further shorten the instrument for more efficient measurement. After the selection based on the item information and content, we obtained a 15-item scale which retained 62.80% of the test information of the D28-C over the D range of (−3, 3). However, it still provided test information of more than 5 across the trait level of (−1, 3), which means that the scale reliability was still acceptable at most levels of D (larger than 0.7) [53]. Given that personality psychological studies often tend to focus on the relationships among different variables, the questionnaire often consists of a number of scales. Thus, the shortened version of the D scale may be more effective in this situation.

## 6. Conclusions

The present study translated the D35 into Chinese and used IRT methods to analyze the Chinese version of the D scale. After removing four items that contributed little to the whole scale, we obtained a 28-item Chinese version of D factor scale (D28-C). This work concluded that the D28-C had acceptable psychometric properties and provided a precise measurement of D from low to very high levels, which means that it can be applied to most students. Although the whole scale can be regarded as invariant across gender, item 13 may require further attention or modification to reduce the gender non-invariance in the context of Chinese culture. In addition, on the basis of the item information and content, a brief form of the D scale was obtained for more effective measurement.

Despite the promising findings, several limitations should be taken into account. First, although the sample size was acceptable, the participation was restricted to college students and less than 30% were men. These facts might limit the generalizability of our findings. Future research should recruit gender-balanced samples with a wide age range, including adolescents and older people. Second, only gender was used to test DIF in the current study. To ensure the generalizability of the scale, other variables, such as age and occupation, could be considered in further studies. Present studies found that individuals scored lower as age increased [10]. Also, it was found that some dark traits were positively related to success in the competitive atmosphere [53]. It is possible that individuals who work in highly competitive jobs, such as athletes, have a higher mean level of some dark traits and a more positive attitude towards these traits. However, these factors were not explored in the current study due to the sample limitation.

## Figures and Tables

**Figure 1 ijerph-19-12787-f001:**
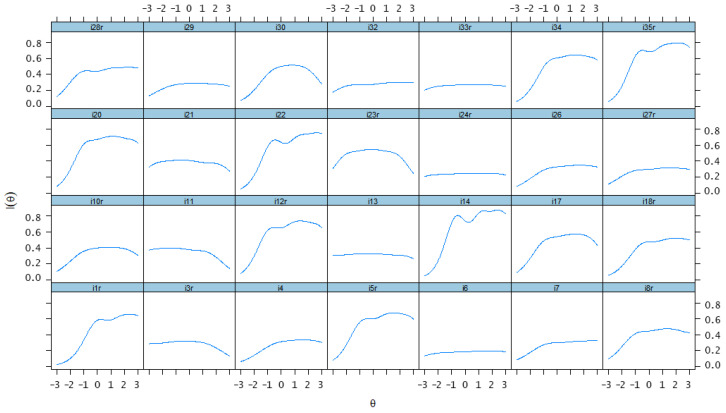
Item information of D28-C (the final Chinese version of D scale). r means that the corresponding item is negatively keyed.

**Figure 2 ijerph-19-12787-f002:**
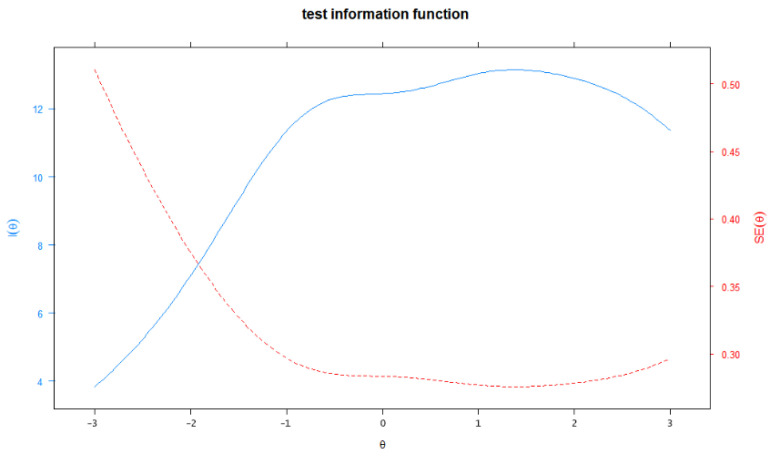
Test information of D28-C.

**Figure 3 ijerph-19-12787-f003:**
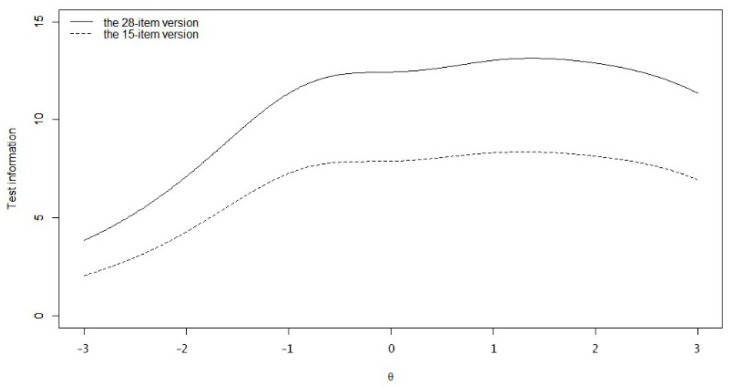
Test information of the initial and final version of D scale.

**Table 1 ijerph-19-12787-t001:** Demographics of Participants (***N*** = 762).

	*N*	Percentage
Gender		
Female	511	67.06%
Male	251	32.94%
Age		
17–19	259	33.99%
20–22	467	61.29%
23–25	34	4.46%
26–27	2	0.26%
Education		
Junior college	19	2.49%
Bachelor’s degree	719	94.36%
Master’s degree or higher	24	3.15%
Total	762	100%

**Table 2 ijerph-19-12787-t002:** Mean scores and standard deviation for the 32-item D scale.

Item	Item Content	Mean	*SD*
I1	It is hard for me to see someone suffering.	2.12	0.89
I2	Payback needs to be quick and nasty.	3.33	1.23
I3	All in all, it is better to be humble and honest than important and dishonest.	2.59	1.26
I4	If I had the opportunity, then I would gladly pay a small sum of money to see a classmate who I do not like fail his or her final exam.	2.41	1.38
I5	Most people are basically good and kind.	2.30	0.94
I6	My own pleasure is all that matters.	2.76	1.12
I7	I’ll say anything to get what I want.	2.11	1.08
I8	Hurting people would make me very uncomfortable.	2.06	0.97
I9	Never tell anyone the real reason you did something unless it is useful to do so.	3.60	1.09
I10	If ever I hurt someone, it was not for my enjoyment.	2.27	1.08
I11	I believe that lying is necessary to maintain a competitive advantage over others.	3.20	1.12
I12	I feel sorry if things I do upset people.	1.91	0.84
I13	A person should use any and all means that are to his advantage, taking care of course, that others do not find out.	3.12	1.22
I14	People who mess with me always regret it.	2.82	1.10
I15	In principle, everyone is worth the same.	2.47	1.29
I16	I cannot imagine how being mean to others could ever be exciting.	2.80	1.32
I17	To make money there are no right and wrong ways anymore. Only easy and hard ways.	2.15	1.15
I18	I don’t want people to be afraid of me or my impulses.	2.19	1.06
I20	I would like to make some people suffer, even if it meant that I would go to hell with them.	2.46	1.29
I21	It’s wise to keep track of information that you can use against people later.	3.69	1.08
I22	’’m not very sympathetic to other people or their problems.	2.18	1.02
I23	It does not give me much pleasure to see my rivals fail.	3.23	1.01
I24	I make a point of trying not to hurt others in pursuit of my goals.	1.92	0.81
I26	Why should I care about other people, when no one cares about me?	2.28	1.16
I27	I avoid humiliating others.	1.90	0.91
I28	Most people deserve respect.	1.66	0.79
I29	Someone who hurts me cannot count on my sympathy.	3.50	1.18
I30	I would be willing to take a punch if it meant that someone I did not like would receive two punches.	2.08	1.21
I32	Success is based on survival of the fittest; I am not concerned about the losers.	2.15	1.04
I33	I do not mind sharing the stage.	2.41	1.06
I34	Doing good deeds serves no purpose; it only makes people poor and lazy.	1.96	0.91
I35	Making people feel bad about themselves does not make me feel any better.	2.15	1.00

**Table 3 ijerph-19-12787-t003:** The relative model-fit results of GRM, GRSM, and GPCM.

	−2LL	AIC	BIC
GRM	62,315.82	62,635.82	63,377.57
GRSM	63,451.00	63,585.01	63,895.62
GPCM	62,702.96	63,022.96	63,764.71

**Table 4 ijerph-19-12787-t004:** Parameter estimates of the initial Chinese version of D scale (D32-C) based on GRM.

Item	Discrimination	Threshold			
	a	*b* _1_	*b* _2_	*b* _3_	*b* _4_
I1	1.27	−1.12	0.93	2.44	4.02
I2	0.55	−4.78	−1.86	0.36	2.47
I3	0.97	−1.33	0.07	1.18	2.91
I4	1.27	−0.63	0.49	1.05	1.99
I5	1.00	−1.79	0.73	2.44	4.16
I6	0.94	−2.28	−0.29	1.30	3.12
I7	1.46	−0.59	0.79	1.75	2.99
I8	1.59	−0.72	0.97	2.07	3.08
I9	0.68	−5.46	−2.37	−0.47	1.93
I10	1.52	−0.89	0.55	1.61	2.94
I11	1.18	−2.50	−0.99	0.25	2.17
I12	1.56	−0.62	1.27	2.54	3.68
I13	1.37	−2.01	−0.62	0.35	1.74
I14	0.91	−2.64	−0.41	1.34	2.96
I15	0.68	−1.47	0.60	1.64	4.01
I16	0.35	−3.91	−0.39	2.06	5.93
I17	1.05	−0.71	1.00	1.93	3.26
I18	1.00	−1.10	1.00	2.03	4.05
I20	1.16	−0.96	0.39	1.24	2.42
I21	1.18	−3.18	−1.73	−0.56	1.27
I22	1.54	−0.88	0.78	1.74	2.97
I23	1.05	−3.39	−1.29	0.42	2.49
I24	1.71	−0.67	1.19	2.55	4.06
I26	1.34	−0.88	0.58	1.57	2.54
I27	1.30	−0.45	1.34	2.53	4.15
I28	1.49	−0.01	1.74	2.87	4.45
I29	1.03	−3.43	−1.28	−0.13	1.28
I30	1.03	−0.40	1.12	1.88	3.06
I32	1.49	−0.80	0.85	1.74	2.91
I33	0.79	−1.98	0.64	2.21	4.53
I34	1.03	−0.72	1.42	3.05	4.70
I35	1.23	−0.91	0.86	2.06	4.07

**Table 5 ijerph-19-12787-t005:** Item fits and parameter estimates of the D28-C based on GRM.

Item	Discrimination	Threshold				Item Fit		
	a	*b* _1_	*b* _2_	*b* _3_	*b* _4_	*S-χ* ^2^	*df*	*p_original_*
I1	1.28	−1.11	0.92	2.43	4.00	134.23	112.00	0.08
I3	0.96	−1.35	0.07	1.20	2.96	148.89	164.00	0.80
I4	1.27	−0.63	0.49	1.05	2.00	168.42	158.00	0.27
I5	0.99	−1.79	0.73	2.45	4.18	150.04	124.00	0.06
I6	0.94	−2.27	−0.29	1.30	3.12	130.50	153.00	0.91
I7	1.44	−0.59	0.80	1.76	3.01	127.24	119.00	0.29
I8	1.62	−0.71	0.97	2.06	3.05	109.36	101.00	0.27
I10	1.53	−0.88	0.55	1.60	2.93	109.92	115.00	0.62
I11	1.16	−2.53	−1.00	0.26	2.19	153.37	150.00	0.41
I12	1.59	−0.61	1.26	2.51	3.65	80.95	82.00	0.51
I13	1.33	−2.04	−0.63	0.36	1.77	151.90	146.00	0.35
I14	0.90	−2.67	−0.41	1.36	3.00	176.46	158.00	0.15
I17	1.05	−0.71	1.00	1.92	3.25	132.42	137.00	0.60
I18	1.01	−1.09	1.00	2.01	4.01	147.41	134.00	0.20
I20	1.14	−0.97	0.40	1.25	2.45	183.17	158.00	0.08
I21	1.14	−3.25	−1.76	−0.57	1.30	143.58	132.00	0.23
I22	1.56	−0.87	0.78	1.73	2.95	143.02	116.00	0.05
I23	1.05	−3.39	−1.29	0.42	2.50	145.78	135.00	0.25
I24	1.76	−0.65	1.17	2.52	4.02	99.51	77.00	0.04
I26	1.35	−0.87	0.58	1.56	2.52	141.70	132.00	0.27
I27	1.31	−0.45	1.34	2.52	4.14	101.61	100.00	0.44
I28	1.48	−0.01	1.75	2.89	4.48	81.00	77.00	0.36
I29	1.01	−3.48	−1.29	−0.13	1.30	191.86	140.00	0.00
I30	1.03	−0.39	1.13	1.88	3.06	154.13	142.00	0.23
I32	1.48	−0.81	0.85	1.75	2.93	132.27	119.00	0.19
I33	0.79	−1.98	0.64	2.22	4.53	166.28	140.00	0.06
I34	1.03	−0.72	1.41	3.04	4.68	94.96	106.00	0.77
I35	1.25	−0.90	0.85	2.03	4.01	115.05	125.00	0.73

**Table 6 ijerph-19-12787-t006:** The results of Differential item functioning (DIF) by gender for D28-C.

Item	Gender		Item	Gender	
	*χ* ^2^	*p_original_*		*χ* ^2^	*p_original_*
I1	0.95	0.33	I20	0.35	0.56
I3	0.08	0.78	I21	0.49	0.48
I4	0.16	0.69	I22	0.23	0.63
I5	0.00	0.99	I23	0.17	0.68
I6	1.22	0.27	I24	0.34	0.56
I7	1.82	0.18	I26	0.20	0.66
I8	0.07	0.80	I27	0.16	0.69
I10	3.92	0.05	I28	1.70	0.19
I11	0.13	0.72	I29	1.81	0.18
I12	8.49	0.00	I30	5.67	0.02
I13	10.13	0.00	I32	0.06	0.80
I14	0.06	0.81	I33	1.01	0.32
I17	0.85	0.36	I34	0.09	0.77
I18	1.64	0.20	I35	1.15	0.28

## Data Availability

All data that are related to the findings of this study are available from the author upon reasonable request.

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
