# Peer review of "Item Response Theory Analysis of the Dark Factor of Personality Scale for College Students in China"

_ijerph, 2022, doi:10.3390/ijerph191912787_

Round 1

Reviewer 1 Report

The authors propose the work of the Dark Factor of Personality in Chinese. Specifically the research aims to translate the Dark Factor of Personality Scale 73 into Chinese and utilise IRT to evaluate the psychometric properties of the Chinese ver-74 sion scale. Moreover, the research also aims to develop a short version of the Chinese D scale to facilitate a quick and 75 efficient assessment. The work is quite interesting as there are indeed cultural differences between Western and Eastern countries,

  • The introduction was written quite comprehensive, however, the background statement could be improved. As far as the reviewer know, there are several works have been done in the Dark Factor of Personality in Chinese. What is the differences between the works and the authors’?
  • Section 3 is too short, it can be combined with section 1 or 2.
  • The methods are comprehensively described and the results are thoroughly discussed.
  • Check the font format of several paragraph in Section 6 
  • The limitation of work and the future work direction can be presented in the conclusion.

Author Response

Dear reviewer, 

Thank you for your comments on our manuscript entitled Item response theory analysis of the dark factor of personality scale for college students in China (ID: ijerph - 1899385).
We have carefully considered the suggestions and made some changes to the manuscript. Responds to the comments are put in the attachment.

Have a nice day!

Yours, sincerely,

Tao Xin

Reviewer 2 Report

The study transferred a scale to measure dark personality traits, D, from a Western to a Chinese context. The method is well explained and motivated and the statistical analysis is extensive and clearly described. Overall, the study answered its research questions, and the main limitations, especially the homogeneous sample, were brought up and discussed.

Minor points, comments section by section:

ABSTRACT

The abbreviation "IRT" is used but not explained - the full term is written in other places in the abstract.

INTRODUCTION

A well-written introduction that nicely balances the negative and the positive aspects of dark traits.
The present study is well-motivated, and put into the context of some, but not much, previous research. Are there no other research on dark traits in Eastern/Chinese contexts that could be included to further contextualize the study?

"IRT" appears on line 67 without any explanation of the abbreviation.

BACKGROUND

The background gives essential details that further motivates the current study.

"Machiavellianism" could be explained, perhaps in the Background.

MATERIALS AND METHODS

How does the sample's e.g., young age affect the results, considering that personality may still be under development at age 17? (Table 1 shows the youngest age as 17 while the text states 18.)

A sample size justification is missing (see, e.g., Lakens, 2022). Although the sample size gives the study sufficient power, a formal power calculation would make this clearer.
Lakens, D. (2022). Sample Size Justification. Collabra: Psychology, 8(1). https://doi.org/10.1525/collabra.33267

"3 items that are difficult to understand and unsuitable in the background of Chinese culture and language custom were deleted": Although space may not allow, given the goal of the present study to adapt the scale to a Chinese context, it would be interesting to know what these items were and why they were problematic.

Were there any counterbalancing/randomization of question order? Even always starting with, e.g., the gender question may bias the participants to answer in a gender-biased way.

Further information about the software packages could be given so that an interested reader could follow up.

Ethical details (e.g., ethical approval), except that a consent form was used, are not given.

RESULTS

Statistical analyses are extensive, clearly written and motivated.

Cohen's d could be supplied with the t-test results.

DISCUSSION

The results are discussed well in relation to culture and methodological limitations.

Author Response

Dear reviewer, 

Thank you for your comments on our manuscript entitled Item response theory analysis of the dark factor of personality scale for college students in China (ID: ijerph - 1899385).
We have carefully considered your kind advice and made corresponding changes to the manuscript. Responds to the comments are put in the attachment.

Have a good day!

Sincerely yours,

Tao, Xin
